# RAINBOW: Haplotype-based genome-wide association study using a novel SNP-set method

**Kosuke Hamazaki**, **Hiroyoshi Iwata** *

Department of Agricultural and Environmental Biology, Graduate School of Agricultural and Life Sciences, The University of Tokyo, Tokyo, Japan

* aiwata@mail.ecc.u-tokyo.ac.jp

## Abstract

Difficulty in detecting rare variants is one of the problems in conventional genome-wide association studies (GWAS). The problem is closely related to the complex gene compositions comprising multiple alleles, such as haplotypes. Several single nucleotide polymorphism (SNP) set approaches have been proposed to solve this problem. These methods, however, have been rarely discussed in connection with haplotypes. In this study, we developed a novel SNP-set method named "RAINBOW" and applied the method to haplotype-based GWAS by regarding a haplotype block as a SNP-set. Combining haplotype block estimation and SNP-set GWAS, haplotype-based GWAS can be conducted without prior information of haplotypes. We prepared 100 datasets of simulated phenotypic data and real marker genotype data of *Oryza sativa* subsp. *indica*, and performed GWAS of the datasets. We compared the power of our method, the conventional single-SNP GWAS, the conventional haplotype-based GWAS, and the conventional SNP-set GWAS. Our proposed method was shown to be superior to these in three aspects: (1) controlling false positives; (2) in detecting causal variants without relying on the linkage disequilibrium if causal variants were genotyped in the dataset; and (3) it showed greater power than the other methods, i.e., it was able to detect causal variants that were not detected by the others, primarily when the causal variants were located very close to each other, and the directions of their effects were opposite. By using the SNP-set approach as in this study, we expect that detecting not only rare variants but also genes with complex mechanisms, such as genes with multiple causal variants, can be realized. RAINBOW was implemented as an R package named "RAINBOWR" and is available from CRAN (https://cran.r-project.org/web/packages/RAINBOWR/index.html) and GitHub (https://github.com/KosukeHamazaki/RAINBOWR).

## Author summary

Detecting rare variants has been one of the most problematic problems in GWAS. Here, we proposed a novel SNP-set GWAS approach, which is superior in controlling false positives and detecting rare variants compared with conventional approaches, and

**Data Availability Statement:** We implemented the method to an R package named RAINBOWR. RAINBOWR is deposited in the CRAN (Comprehensive R Archive Network), https://cran.r-project.org/web/packages/RAINBOWR/index.html,

and in the "KosukeHamazaki/RAINBOWR" repository in the GitHub, https://github.com/KosukeHamazaki/RAINBOW. The datasets and scripts generated and analyzed during the current study are available from the "KosukeHamazaki/HGRAINBOW"repository in the GitHub,https://github.com/KosukeHamazaki/HGRAINBOW.

**Funding:** This work was supported by JST CREST (https://www.jst.go.jp/kisoken/crest/en/index.html) Grant Number JPMJCR16O2, Japan. The funders had no role in study design, data collection, and analysis, decision to publish, or preparation of the manuscript.

**Competing interests:** The authors have declared that no competing interests exist.

implemented this method as an R package named "RAINBOWR" (Reliable Association INference By Optimizing Weights with R). In this article, we introduce the application of RAINBOW to haplotype-based GWAS by regarding a haplotype block as a SNP-set, which enables one to perform haplotype-based GWAS without prior haplotype information. We showed that the haplotype-based GWAS with the RAINBOW package succeeded in detecting causal variants with complex mechanisms that were not detected by any other conventional methods. RAINBOW also offers a fast single-SNP GWAS method. RAINBOW offers not only a SNP-set GWAS that can be applied to universal situations but also one that is faster with the restircted situations using linear kernel for constructing the Gram matrix of SNP-set of interest. We also used Rcpp (functions for using C++ in R) for the RAINBOW implementation to achieve faster computation. We believe that our package will lead to the detection of novel genes associated with biologically and agronomically essential traits.

This is a *PLOS Computational Biology* Software paper.

## Introduction

With the decreasing cost and increasing throughput of next-generation sequencing, the number of accessions that can be used for genome-wide association study (GWAS) is increasing [1–3]. Using such large sequencing data, GWAS is now widely used not only in human but also in plant and animal genetics and breeding, and has identified novel genes related to important agronomic traits [4–6]. One example of large next-generation sequencing data is that of the "3,000 rice genomes project" as used in this study [7, 8], data from which are available in the "Rice SNP-Seek Database" [9–11]. GWAS results using these data have already been reported [12].

Despite the enhancement of such public data, the conventional GWAS method still faces obstacles in the detection of unknown candidate genes. One common example is its difficulty in detecting rare alleles or rare variants. One problem caused by rare variants is that the non-causal markers that have a strong linkage disequilibrium (LD) with one causal rare variant indicate a higher detection power than the true causal rare variant, which may interfere with the detection of the true causal variant. This phenomenon is known as "synthetic association", and often happens when the minor allele frequency (MAF) of the non-causal marker is higher than that of the true rare variant [13]. This problem is closely related to the complex gene compositions comprising multiple alleles such as haplotypes because genes related to important agronomic traits often consist of multiple rare alleles, and this is why haplotypes are hard to detect using GWAS [14].

Several methods have been proposed to solve this problem. The sequence kernel association test (SKAT) is one of the methods used to detect rare variants, and has been used mainly in human genomics [15]. The SKAT employs a single nucleotide polymorphism (SNP) set approach, which tests multiple SNPs in each SNP-set at the same time. The SKAT evaluates the significance of the variance explained by a SNP-set of interest as a random effect using a mixed effect model approach [16, 17]. The fatal drawback of the original SKAT is that the model does not take the effects of family relatedness into account as a random effect, which results in false positives for GWAS in materials with a strong population structure or family relatedness, such as in the world collection of rice germplasm used in this study. Several methods were also proposed to overcome another SKAT drawback: a weighting scheme of the

SKAT for rare and common variants can lead to loss of power of common variants, but their models also do not include the term for correcting the confounding effects of family relatedness [18, 19].

To solve the fatal drawbacks of the original SKAT, several methods whose models include the term of family relatedness as random effects to control false positives have been previously proposed [20–22]. From a statistical point of view, these methods usually perform the score test [23], which is a computationally efficient method since it requires variance component estimation only for the null model. In terms of the detection power, however, the score test is not necessarily the best method for testing the random effects in the mixed effects model [24]. The likelihood-ratio (LR) test [25, 26] is another candidate used to test the variance of a SNP-set of interest, and several methods have been proposed that use the LR test for SNP-set GWAS in family samples [24, 27]. In particular, Lippert *et al.* implemented a computationally efficient SNP-set GWAS method using the LR test, and reported that the LR test showed greater power than the score test [24]. Despite being such an efficient method, Lippert *et al.* mainly used a linear kernel for constructing the Gram matrix from each SNP-set, and therefore other kernels, such as a Gaussian kernel or an exponential kernel, cannot be used for constructing the Gram matrix in their method.

Haplotype-based approaches, which try to improve the detection power of causal haplotypes, make sense from the point of view that a gene functions as one gene set, not as each SNP in the gene set. These haplotype-based approaches are expected to control false positives better than the single-SNP method because the haplotype-based methods focus on the entire haplotype block, not on each SNP in the haplotype block. These methods are also expected to reveal the complex mechanism of causal haplotypes that cannot be detected when focusing on one SNP, such as repulsion states between two causal quantitative trait loci (QTL) located close to each other. However, only a few methods for haplotype-based GWAS have so far been proposed. In plant genomics, Yano *et al.* performed a haplotype-based GWAS by testing the effects of haplotypes while regarding dummy variables of haplotype groups as fixed effects, and found new candidate genes related to heading date for rice [28]. Other approaches have been proposed in animal genomics, which estimated ancestral haplotype effects by regarding them as random effects [29, 30]. In their methods, each pairwise element of a covariance matrix for the random effects was determined as 1 if individuals belong to the same ancestral haplotype, and 0 if otherwise. However, these conventional haplotype-based GWAS methods require haplotype information a priori, and it is not so easy to apply these methods at the genome-wide level.

In this study, we extended the multi-kernel mixed effects model more generally to take family relatedness into account, while enabling computational speed-up for some limited cases, and developed a novel SNP-set GWAS approach named RAINBOW (Reliable Association INference By Optimizing Weights). We also estimated haplotype blocks from genome-wide marker genotype data, and used them as SNP-sets for analysis with RAINBOW to enable haplotype-based GWAS without prior haplotype information.

## Materials and methods

All statistical analyses in this study were conducted using R version 3.6.0 [31], and figures were produced using the R package ggplot2 version 3.2.1 [32]. Our R package, RAINBOWR, was implemented using the R packages Rcpp version 1.0.2 [33–35] and RcppEigen version 0.3.3.5.0 [36] to reduce the computational time required for solving the multi-kernel mixed-effects model described below. The overall simulation framework in this study is shown in S1 Fig as a flow chart.

## Methods for RAINBOW

In this subsection, we describe the basic idea of RAINBOW.

**RAINBOW model.** The RAINBOW model can be written as

$$\mathbf{y} = \mathbf{X}\boldsymbol{\beta} + \mathbf{Z}_{\mathrm{c}}\mathbf{u}_{\mathrm{c}} + \mathbf{Z}_{\mathrm{r}_i}\mathbf{u}_{\mathrm{r}_i} + \epsilon, \tag{1}$$

where $\mathbf{y}$ is a $n \times 1$ vector of phenotypic values, $\mathbf{X}\boldsymbol{\beta}$ is a $n \times 1$ vector of fixed effects including an intercept, a term to correct the population structure and other covariates, $\mathbf{Z}_{\mathrm{c}}\mathbf{u}_{\mathrm{c}}$ and $\mathbf{Z}_{\mathrm{r}_i}\mathbf{u}_{\mathrm{r}_i}$ are $n \times 1$ vectors of random effects, and $\epsilon$ is a $n \times 1$ vector of residual errors. Here $\boldsymbol{\beta}$ is a $p \times 1$ vector of fixed effects, where $p$ is the number of fixed effects. $\mathbf{u}_{\mathrm{c}}$ and $\mathbf{u}_{\mathrm{r}_i}$ are $m_{\mathrm{c}} \times 1$ and $m_{\mathrm{r}_i} \times 1$ vector of genotypic values respectively, where $m_{\mathrm{c}}$ is the number of genotypes for additive polygenetic effects and $m_{\mathrm{r}_i}$ is the number of genotypes for $i$-th SNP-set of interest. $\mathbf{X}$, $\mathbf{Z}_{\mathrm{c}}$ and $\mathbf{Z}_{\mathrm{r}_i}$ are $n \times p$, $n \times m_{\mathrm{c}}$ and $n \times m_{\mathrm{r}_i}$ design matrices that correspond to $\boldsymbol{\beta}$, $\mathbf{u}_{\mathrm{c}}$ and $\mathbf{u}_{\mathrm{r}_i}$ respectively. As the following formula Eq 2, we assume that the polygenetic effect $\mathbf{u}_{\mathrm{c}}$ follows the multivariate normal distribution whose variance-covariance matrix is proportional to the additive numerator relationship matrix $\mathbf{K}_{\mathrm{c}}$.

$$\mathbf{u}_{\mathrm{c}} \sim \mathrm{MVN}(\mathbf{0}, \mathbf{K}_{\mathrm{c}}\sigma_{\mathrm{c}}^2), \tag{2}$$

where $\sigma_{\mathrm{c}}^2$ is the additive genetic variance to be estimated in the "Estimation of variance components" section, and here $m_{\mathrm{c}} \times m_{\mathrm{c}}$ matrix $\mathbf{K}_{\mathrm{c}} = \mathbf{A}$, where $\mathbf{A}$ is the known additive genetic relationship matrix estimated from marker genotype data $\mathbf{W}_{\mathrm{c}}$ [37].

We also assume that the random effects from $i$-th SNP-set of interest $\mathbf{u}_{\mathrm{r}_i}$ follows the multivariate normal distribution whose variance-covariance matrix is proportional to the Gram matrix $\mathbf{K}_{\mathrm{r}_i}$.

$$\mathbf{u}_{\mathrm{r}_i} \sim \mathrm{MVN}(\mathbf{0}, \mathbf{K}_{\mathrm{r}_i}\sigma_{\mathrm{r}_i}^2), \tag{3}$$

where $\sigma_{\mathrm{r}_i}^2$ is the genetic variance for $i$-th SNP-set to be estimated in the "Estimation of variance components" section, and $\mathbf{K}_{\mathrm{r}_i}$ is the known $m_{\mathrm{r}_i} \times m_{\mathrm{r}_i}$ Gram matrix estimated from marker genotype data $\mathbf{W}_{\mathrm{r}_i}$ belonging to the $i$-th SNP-set. We offer a linear, an exponential and a Gaussian kernel for the Gram matrix $\mathbf{K}_{\mathrm{r}_i}$, and faster computation can be realized for the linear kernel case (Supplementary Note in S1 Appendix) [24].

Finally, the residual term is assumed to identically and independently follow a normal distribution as shown in the following equation.

$$\epsilon \sim \mathrm{MVN}(\mathbf{0}, \mathbf{I}_n\sigma_{\mathrm{e}}^2), \tag{4}$$

where $\mathbf{I}_n$ is a $n \times n$ identity matrix and $\sigma_{\mathrm{e}}^2$ is estimated in the "Estimation of variance components" section.

**Estimation of variance components.** The variance components were estimated by maximum-likelihood (ML) [26, 38] and restricted maximum-likelihood (REML) [39]. Here we explain how to obtain ML and REML estimates of Eq 1 for the general $\mathbf{K}_{\mathrm{r}_i}$.

First we estimated the weights (we define $w_{\mathrm{c}}$ and $w_{\mathrm{r}_i}$) between the genetic variances ($\sigma_{\mathrm{c}}^2$ and $\sigma_{\mathrm{r}_i}^2$) by the following algorithm.

1. Setting initial parameters for $w_{\mathrm{c}}$ and $w_{\mathrm{r}_i}$:

$$w_{\mathrm{c}} = w_{\mathrm{r}_i} = \frac{1}{2}. \tag{5}$$

2. Computing the following $n \times n$ matrix $\mathbf{K_s}$:

$$\mathbf{K_s} = \mathbf{Z_c K_c Z_c^T} w_c + \mathbf{Z_{r_i} K_{r_i} Z_{r_i}^T} w_{r_i}. \tag{6}$$

3. Solving the following single-kernel linear mixed model (LMM) by using EMMA (efficient mixed model association) or GEMMA (genome-wide efficient mixed model association) [40, 41].

$$\mathbf{y} = \mathbf{X}\boldsymbol{\beta} + \mathbf{u_s} + \epsilon, \tag{7}$$

where

$$\mathbf{u_s} \sim \mathrm{MVN}(\mathbf{0}, \mathbf{K_s}\sigma_s^2.) \tag{8}$$

4. Computing the full log likelihood ($l_\mathrm{F}$) or the restricted log likelihood ($l_\mathrm{R}$) of Eq 7 by using estimated parameters; $\hat{\beta}$, $\hat{\sigma}_s^2$ and $\hat{\sigma}_e^2$:

$$
\begin{aligned}
l_\mathrm{F}(\mathbf{y}; \hat{\boldsymbol{\beta}}, \hat{\sigma}_s, \hat{\delta}) &= \frac{1}{2}\Big[ -n\log(2\pi\hat{\sigma}_s^2) - \log|\hat{\mathbf{H}}| \\
&\quad -\frac{1}{\hat{\sigma}_s^2}(\mathbf{y} - \mathbf{X}\hat{\boldsymbol{\beta}})^\mathrm{T} \hat{\mathbf{H}}^{-1}(\mathbf{y} - \mathbf{X}\hat{\boldsymbol{\beta}})\Big],
\end{aligned} \tag{9}
$$

$$
\begin{aligned}
l_\mathrm{R}(\mathbf{y}; \hat{\sigma}_s, \hat{\delta}) &= l_\mathrm{F}(\mathbf{y}; \hat{\boldsymbol{\beta}}, \hat{\sigma}_s, \hat{\delta}) \\
&\quad +\frac{1}{2}[p\log(2\pi\hat{\sigma}_s^2) + \log|\mathbf{X^T X}| \\
&\quad -\log|\mathbf{X^T \hat{H}^{-1} X}|].
\end{aligned} \tag{10}
$$

Here $\hat{\mathbf{H}}$ is

$$\hat{\mathbf{H}} = \frac{\hat{\mathbf{V}}}{\hat{\sigma}_s^2} = \mathbf{K_s} + \hat{\delta}\mathbf{I}_n. \tag{11}$$

where $\hat{\mathbf{V}}$ is a phenotypic variance-covariance matrix and $\hat{\delta} = \hat{\sigma}_e^2 / \hat{\sigma}_s^2$.

5. Optimizing $w_c$ and $w_{r_i}$ over maximization of the full/restricted log likelihood by using L-BFGS optimization method through repeating step 2-4 [42].

After estimating the weights $w_c$ and $w_{r_i}$, we estimated the variance components ($\sigma_s^2$ and $\sigma_e^2$) of the model Eqs 7 and 8 by EMMA/GEMMA using $\hat{w}_c$ and $\hat{w}_{r_i}$. Then we obtained $\hat{\sigma}_c^2 = \hat{w}_c\hat{\sigma}_s^2$ and $\hat{\sigma}_{r_i}^2 = \hat{w}_{r_i}\hat{\sigma}_s^2$.

Our fitting method, as described above, is a two-step approach, which first estimates the weights of genetic variances, and then estimates the variance components of the model shown in Eqs 7 and 8 by EMMA/GEMMA with the estimated weights. On the other hand, some fitting methods that directly estimate the variance components for Eq 1 via AIREML (average information REML) [43] have also been proposed and implemented in some packages/software [44, 45]. The advantage of our two-step approach compared with the direct estimation approach via AIREML is that the search space of the weights is limited to the interval [0, 1], and the convergence is relatively warranted [46] even when the heritability is too low/high.

**Likelihood ratio test for GWAS.** To test the significance of each SNP-set, we performed the LR test of whether $\sigma^2_{r_i} = 0$ or not. As a null hypothesis, the following model, which does not include the term of SNP-set effects was assumed.

$$\mathbf{y} = \mathbf{X}\boldsymbol{\beta} + \mathbf{Z}_c\mathbf{u}_c + \epsilon. \tag{12}$$

In contrast, as an alternative hypothesis model, the multi-kernel linear mixed model (MKLMM) of Eq 1 was assumed. Therefore, we computed the following deviance after the estimation of variance components for each SNP-set.

$$D = 2 \times (\hat{l}_{R,model} - \hat{l}_{R,null}), \tag{13}$$

where $\hat{l}_{R,model}$ is the maximum of the restricted log likelihood for the model of Eq 1 and $\hat{l}_{R,null}$ is the maximum of the restricted log likelihood for the model of Eq 12.

Finally, we tested the significance of $\sigma^2_{r_i}$ and calculate the $p$-value by assuming that the deviance in Eq 13 followed the mixture of two chi-square distributions with different degrees of freedom [47, 48].

$$D \sim \pi_0\chi^2_0 + (1 - \pi_0)\chi^2_1, \tag{14}$$

where $\pi_0$ is the mixture parameter and here we used $\pi_0 = 1/2$.

## Materials and simulations

**Genotype data.** In this study, 414 accessions of *Oryza sativa* subsp. *indica* were collected from "the 3,000 rice genomes project" (S1 Table) [7]. We used a marker genotype consisting of core SNPs defined by the Rice SNP-Seek Database as "404k CoreSNP Dataset". Imputations were imputed using Beagle version 5.0 [49, 50]. We analyzed only bi-allelic sites over all accessions with a MAF $\geq 0.025$ by using VCFtools version 0.1.15 [51]. In the following analysis, genotypes are represented as -1 (homozygous of the reference allele), 1 (homozygous of the alternative allele) or 0 (heterozygous of the reference and alternative alleles). As a result of this data processing, marker genotypes with 112,630 SNPs were used for the following simulation study.

**Estimation of haplotype block.** To perform haplotype-based GWAS by regarding each haplotype block as a SNP-set, haplotype blocks were estimated from marker genotype data by using PLINK 1.9 [52–54]. As a result of estimation, we obtained 15,275 haplotype blocks consisting of 78,237 SNPs.

**Simulation of phenotype data.** We considered two scenarios to validate our novel haplotype-based GWAS approach. In both models, phenotypic values were simulated as follows.

$$\mathbf{y} = \mathbf{X}_1\beta_1 + \mathbf{X}_2\beta_2 + \mathbf{X}_3\beta_3 + \mathbf{Z}\mathbf{u} + \mathbf{e}, \tag{15}$$

where $\mathbf{y}$ is the vector of simulated phenotypic values of 414 accessions, $\mathbf{X}_1$, $\mathbf{X}_2$ and $\mathbf{X}_3$ correspond to three quantitative trait nucleotides (QTNs) scored as -1, 0 or 1 (hereinafter, referred to as "QTN1", "QTN2" and "QTN3" respectively), $\beta_1$, $\beta_2$ and $\beta_3$ are scalars representing the effects of the three QTNs, $\mathbf{u}$ is the vector of polygenetic effects, and $\mathbf{e}$ is the vector of the residuals.

Here, QTN1 and QTN2 were randomly selected from all genome-wide SNPs to satisfy that they belonged to the same haplotype block that harbored more than 4 SNPs. QTN3 was randomly selected from all the SNPs. We assumed that the effects of QTN1 and QTN2 had a variance 4 times greater than that of the effects of QTN3 to mainly check the detection power for the haplotype block. More details about the other terms are described in S1 Appendix.

The difference between two scenarios is based on the directions of the two QTN effects $\beta_1$ and $\beta_2$. Scenario 1 assumed that the directions of two effects were identical. That is,

$$\beta_1 = \begin{cases} \beta_2 & (\rho_{12} \geq 0) \\ -\beta_2 & (\rho_{12} < 0) \end{cases}, \tag{16}$$

where $\rho_{12}$ is Pearson's correlation coefficient between $\mathbf{X}_1$ and $\mathbf{X}_2$. We call this model as "coupling".

Conversely, scenario 2 assumed that the directions of the two effects were opposite. That is,

$$\beta_1 = \begin{cases} -\beta_2 & (\rho_{12} \geq 0) \\ \beta_2 & (\rho_{12} < 0) \end{cases} \tag{17}$$

We call this scenario 2 as "repulsion".

## Evaluation of RAINBOW

**Comparison of four methods.** To validate our novel approach, we compared the following four methods: a single-SNP GWAS [55], a haplotype-based GWAS introduced by Yano *et al.* (hereinafter, referred to as "HGF") [28], the SKAT [15] as a SNP-set approach, and our novel approach, RAINBOW. For all methods, to account for the population structure, the two eigen vectors (which correspond to the top two eigen values) of the additive genetic relationship matrix were included in the model as fixed effects. The details of these four methods are described in S1 Appendix.

**Evaluation of the simulation results.** The value of $-\log_{10}(p)$ of each marker or haplotype block was calculated by the four GWAS methods 100 times for the two simulated scenarios, coupling and repulsion. In this study, the following summary statistics were used to evaluate the simulation results.

$-\log_{10}(p)$ **and** $-\log_{10}(p_a)$. The first summary statistic is $-\log_{10}(p)$ of each causal SNP or haplotype block itself. For haplotype-based GWAS methods, HGF, SKAT and RAINBOW, the significance of $\beta_1$ and $\beta_2$ was represented by $-\log_{10}(p)$ of the causal haplotype block to which $\mathbf{X}_1$ and $\mathbf{X}_2$ belong. In the single-SNP GWAS method, the $-\log_{10}(p)$ of $\beta_1$ and $\beta_2$ were calculated separately, even though these SNPs were in the same haplotype. To compare the single-SNP GWAS method with the haplotype-based GWAS methods, the $-\log_{10}(p)$ values were averaged over $\beta_1$ and $\beta_2$.

As some of these methods showed the results of inflated $-\log_{10}(p)$, we defined the following summary statistic to evaluate the degree of inflation.

$$inflator = \frac{1}{L}\sum_{l=1}^{L}(-\log_{10}(p_{false,l})), \tag{18}$$

where $p_{false,l}$ is the $l^{th}$ $p$-values for false positives arranged in increasing order. In this study, $L$ was set as 10. Then we adjusted $-\log_{10}(p)$ of the causal by using the inflator (Eq 18) as follows.

$$-\log_{10}(p_a) = -\log_{10}(p) - inflator, \tag{19}$$

where $p_a$ is the $p$-value adjusted by the inflator.

Here, we calculated each summary statistic in two ways. The first method is to calculate each summary statistic by directly using $-\log_{10}(p)$ of each causal SNP / haplotype block. The other method is to calculate the summary statistics by regarding multiple SNPs or haplotype blocks within the extent of the LD as one set. In this study, we defined SNPs or haplotype

blocks that satisfy the condition that they are within 300 kb from the focused SNP or haplotype block and the condition that their square of the correlation coefficients with the focused SNP or haplotype block are 0.35 or more as one set considering the LD. The highest value of $-\log_{10}(p)$ in the LD region was assumed to represent the values of the SNPs or haplotype blocks within the extent of the LD.

**Recall, precision and F-measure.** We calculated the recall, precision and F-measure means as other summary statistics to evaluate the GWAS results. These summary statistics were calculated from the numbers of SNPs or haplotype blocks that were true positives, false positives, false negatives and true negatives. Here, we regarded a SNP or haplotype block as "positive" when that SNP or haplotype block exceeded the threshold. In this study, the value of $-\log_{10}(p)$ so that the FDR (false discovery rate) was 0.01 was set as the threshold by using the Benjamini-Hochberg method [56, 57]. In addition, these three summary statistics, recall, precision and F-measure, were calculated by assuming that the highest value of $-\log_{10}(p)$ in the LD region represented the values of the SNPs or haplotype blocks within the extent of the LD.

Therefore, recall represents the proportion of causals detected by GWAS. In contrast, precision represents the ratio of the detected SNPs or haplotype blocks that were causals. Finally, F-measure was calculated as the harmonic mean of the recall and the precision, which evaluates the GWAS results comprehensively. The greater these three summary statistics, the better the results of GWAS are. Here we simply took the average of each summary statistic from all the 100 simulation results.

**AUC for regions around causals.** We calculated the mean of the AUC (area under the curve) for regions around the causals as a summary statistic. AUC refers to the area under the ROC (receiver operating characteristic) curve obtained by plotting the false positive rate on the horizontal axis and the true positive rate on the vertical axis when the threshold is varied. In this study, the AUC was calculated for the SNPs or haplotype blocks near the causal SNP / haplotype block (QTN1 and QTN2). In other words, the non-causal markers that had a strong LD with the causal SNP / haplotype block were regarded as false positives under this summary statistic. Therefore, this summary statistic indicates the extent to which the causal itself can be detected by GWAS without relying on the LD. Here, when taking the average of the AUCs obtained from the simulation results, two methods were used, either using all the 100 results or only using the results whose QTN1 and QTN2 were "detected". Here, QTN1 and QTN2 were regarded as "detected" if $-\log_{10}(p_a) \geq 1.5$ for each method.

## Availability of data and material

RAINBOW was implemented as an R package named "RAINBOWR", which offers the single-SNP GWAS method [41, 55] and a novel SNP-set method that includes faster computation for the linear kernel [24]. A stable version of RAINBOWR is available from the CRAN (Comprehensive R Archive Network), https://cran.r-project.org/web/packages/RAINBOWR/index.html. The latest version of RAINBOWR is also available from the "KosukeHamazaki/RAINBOWR" repository in the GitHub, https://github.com/KosukeHamazaki/RAINBOWR. Source codes for the R package RAINBOWR are deposited in S1 File. The datasets generated and analyzed during the current study and their source codes are also available from the "KosukeHamazaki/HGRAINBOW" repository in the GitHub, https://github.com/KosukeHamazaki/HGRAINBOW.

## Results

### The detection power of four methods

The detection power of the four methods was evaluated by the value of $-\log_{10}(p)$ and $-\log_{10}(p_a)$ of QTN1 and QTN2 for the two models, coupling and repulsion (Fig 1). RAINBOW

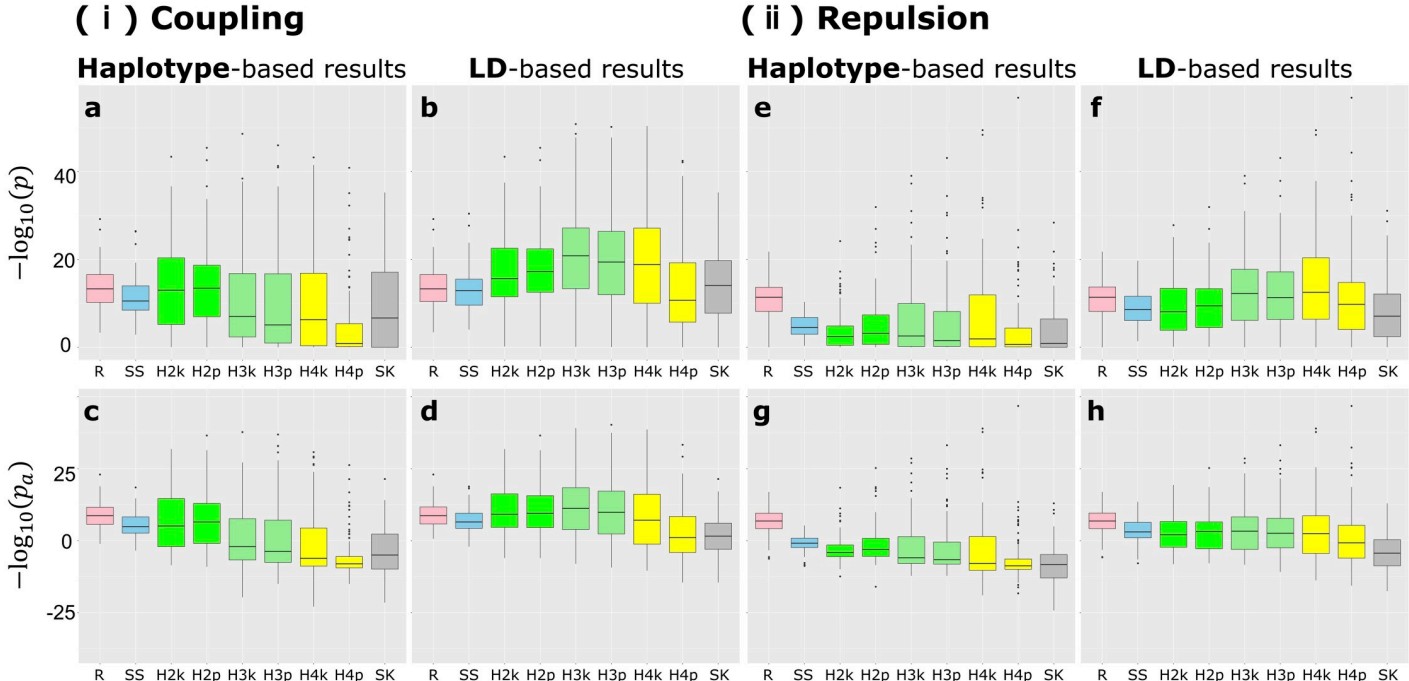

**Fig 1. The detection power of each GWAS method.** Boxplot of the detection power evaluated by $-\log_{10}(p)$ and $-\log_{10}(p_a)$. **a-d**: The results for the "coupling" scenario. **e-h**: The results for the "repulsion" scenario. **a,b,e,f**: The results evaluated by $-\log_{10}(p)$ with the scale on the vertical axis aligned in these four figures. **c,d,g,h**: The results evaluated by $-\log_{10}(p_a)$ with the scale on the vertical axis aligned in these four figures. **a,c,e,g**: The results evaluated by the unit of the causal SNP or haplotype block itself. **b,d,f,h**: The results evaluated by the unit of the regions within the extent of LD. The abbreviation of each method is as follows. **R**: RAINBOW. **SS**: Single-SNP GWAS. **H2k-H4p**: HGF methods. The numbers in the method names correspond to the numbers of the groups they assume. The last letters of the methods are "k" or "p". "k" corresponds to the k-medoids method and "p" corresponds to UPGMA method for the grouping method. **SK**: SKAT.

outperformed the other methods when the significance was evaluated by the causal itself (Fig 1a, 1c, 1e and 1g). However, when the significance was evaluated by the highest values of SNPs or haplotypes within the extent of the LD, other methods, e.g., HGF (k = 2, k-medoids method), showed a greater detection power than RAINBOW (Fig 1b). When the detection power was evaluated by taking the extent of inflation into account, RAINBOW showed as great a power as HGF (k = 2, 3) even if the significance was evaluated by the unit of the LD block (Fig 1d). Moreover, although the detection power of all the GWAS methods for the repulsion scenario was less than that for the coupling scenario, the tendency for RAINBOW to outperform the other methods was clearer for the repulsion scenario than the coupling scenario (Fig 1). Finally, as compared with the other haplotype-based GWAS methods, RAINBOW showed smaller variation among iterations, indicating that the causal variants can be stably detected (Fig 1).

The detection power for QTN3 was also evaluated. The single-SNP GWAS method showed a greater power than RAINBOW when evaluated by $-\log_{10}(p)$ (a,b,e,f in S2 Fig). However, if the detection power was evaluated by $-\log_{10}(p_a)$, RAINBOW showed as great a power as single-SNP GWAS (c,d,g,h in S2 Fig). Contrary to the results for QTN1 and QTN2, the detection power of all the GWAS methods for the repulsion scenario was greater than for the coupling scenario (S2 Fig).

## Recall, precision and F-measure

The characteristics of each GWAS method were evaluated by the recall, precision and F-measure means (Fig 2). For the mean of recall, the HGF methods and SKAT showed higher values

## (a) Coupling

## (b) Repulsion

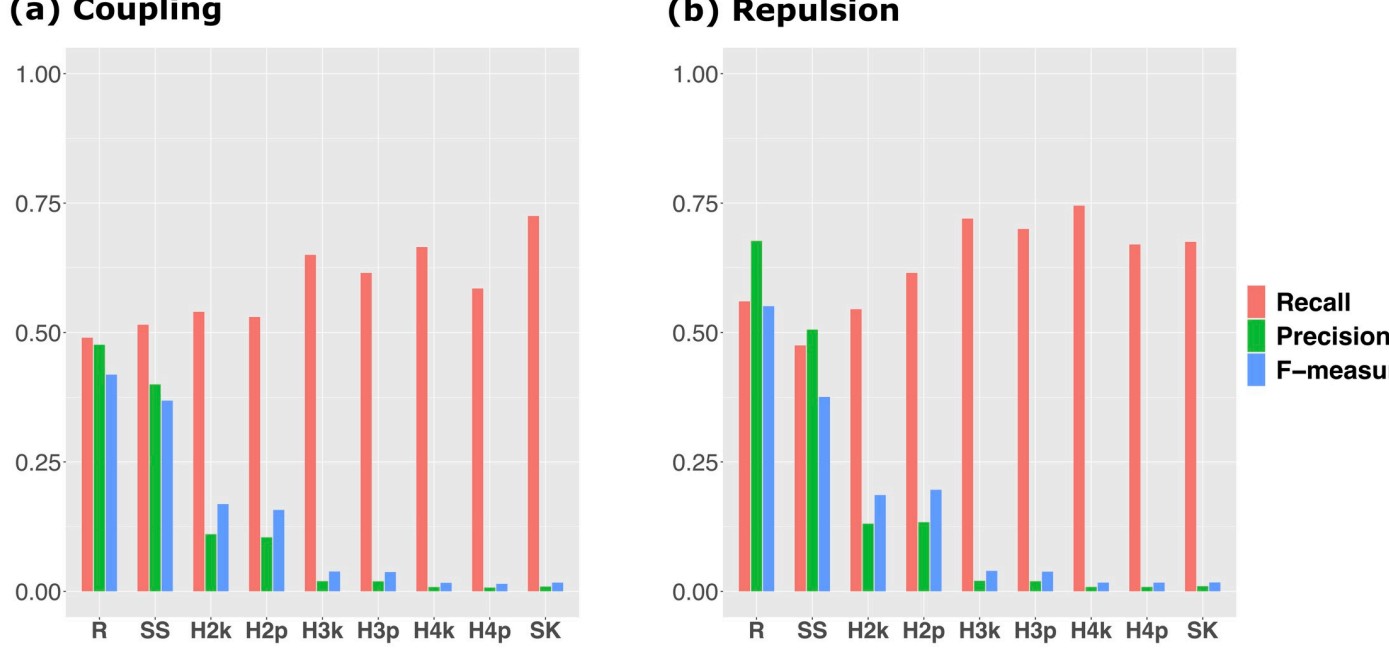

**Fig 2. Recall, precision and F-measure of each GWAS method.** Bar plot of the mean of each summary statistic for 100 simulation results. The red bars show the results for recall, the green bars show the results for precision, and the blue bars show the results for F-measure. **a**: Results for the coupling scenario. **b**: Results for the repulsion scenario. The abbreviations of each method are the same as those of Fig 1.

than RAINBOW and single-SNP GWAS for both scenarios (Fig 2). However, the haplotype-based GWAS methods other than RAINBOW showed low precision. That is, these methods may cause too many false positives. In contrast, RAINBOW and single-SNP GWAS showed higher precision than the remainders, and RAINBOW showed the highest precision among all the scenarios. From the results for the three summary statistics, RAINBOW also showed the highest value for F-measure among the methods. In particular, for the repulsion scenario, the recall of RAINBOW was also higher than that of single-SNP GWAS, which resulted in the large difference of F-measure between these two methods (Fig 2b). A similar tendency was also confirmed when changing the criterion of how to determine the threshold for the Bonferroni's correction [58] for the significance level $\alpha$ = 0.01 (S3 Fig).

To compare the three summary statistics of the two scenarios in more detail, these values for each QTN were also calculated (S4 Fig). For both scenarios, RAINBOW showed the highest recall for QTN1 and QTN2 among the methods (a, b in S4 Fig). In particular, RAINBOW outperformed the other methods in all summary statistics for QTN1 and QTN2 for the repulsion scenario. However, it showed lower recall for QTN3 than the other methods (c, d in S4 Fig). In particular, the recall of RAINBOW for QTN3 was 0 for the coupling scenario (c in S4 Fig). In addition, the three summary statistics of QTN3 for the repulsion scenario were greater than those for the coupling scenario in almost all the methods (c, d in S4 Fig). Regarding these results, a similar trend was confirmed even when changing the criterion of how to determine the threshold for the Bonferroni's correction for the significance level $\alpha$ = 0.01 (S5 Fig).

## AUC for regions around causals

To evaluate how the causal itself can be detected by GWAS without relying on the LD, the AUC means for regions around the causals (QTN1 and QTN2) were compared (Fig 3). The

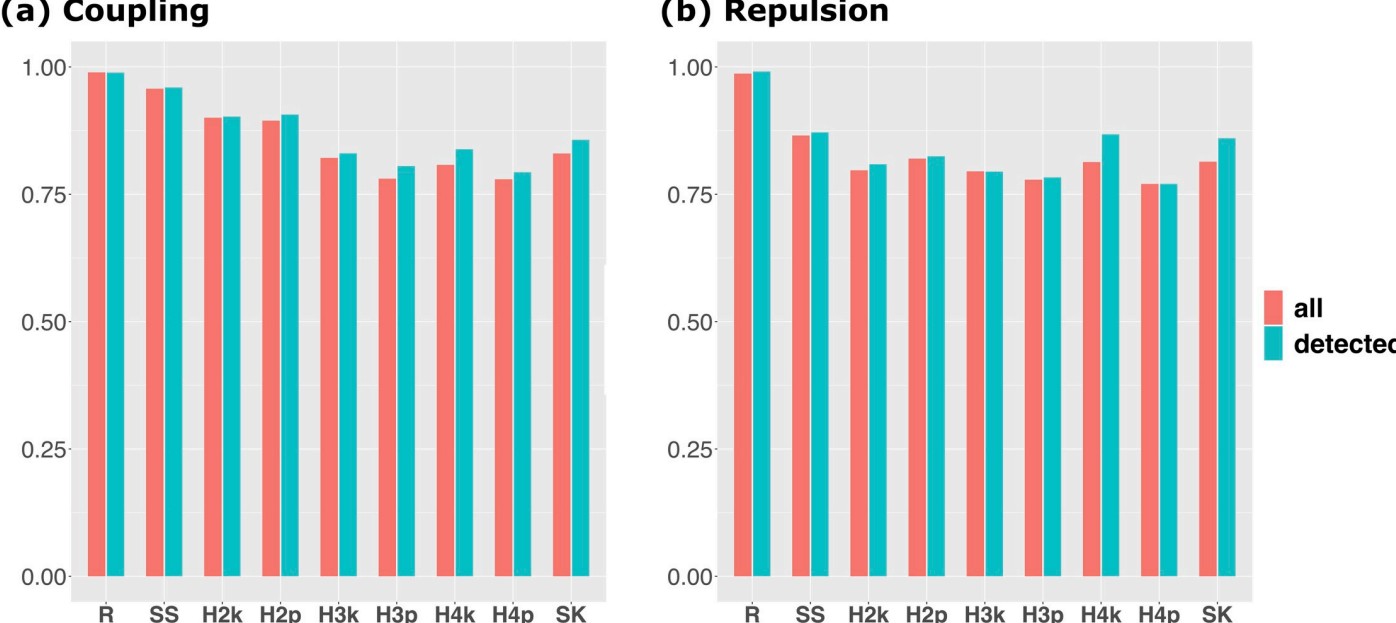

**Fig 3. AUC for regions around causals.** Bar plot of the mean of AUC for regions around causals. This summary statistic indicates the extent to which the causal itself can be detected by GWAS without relying on the LD. The red bars show the results for the means of 100 simulation results and the blue bars show the results for the means of the simulation results whose QTN1 and QTN2 were detected. **a**: Results for the coupling scenario. **b**: Results for the repulsion scenario. The abbreviations of each method are the same as those of Fig 1.

mean of AUC was almost the same when using all simulation results or using only the cases in which QTN1 and QTN2 were detected, although the value of the latter was slightly larger than that of the former in some methods. The results show that RAINBOW outperformed the other methods in both models (Fig 3). Especially, the AUC mean of the single-SNP GWAS method in the repulsion scenario was much smaller than that in the coupling scenario, while RAINBOW was able to maintain a high AUC even in the repulsion scenario (Fig 3b).

## Examples in the repulsion scenario

Of the 100 simulations for the repulsion scenario, there were 7 cases in which QTN1 and QTN2 were detected only by RAINBOW. These cases were selected to satisfy three conditions that $-\log_{10}(p_{a,\mathrm{R}}) \geq 1.5$, $-\log_{10}(p_{a,\mathrm{O}}) \leq 1.2$ and the recall for QTN1 and QTN2 equals to 1. Here, $p_{a,\mathrm{R}}$ represents the adjusted $p$-value of RAINBOW and $p_{a,\mathrm{O}}$ represents the adjusted $p$-value of all the other methods. Although the same analysis was done for the other methods, no method satisfied the three conditions described above. One example of these cases (iteration 48) was shown by comparing the four GWAS methods; RAINBOW, single-SNP GWAS, HGF (the number of groups is 2, the grouping method is UPGMA) and SKAT (Fig 4). The Manhattan plot shows that RAINBOW succeeded in detecting the causal haplotype block (of QTN1 and QTN2) that was not detected by the other methods. Although both QTN1 and QTN2 were also detected by the single-SNP method in one case (iteration 85), the same trend as the results for iteration 48 was seen for the remaining five results (S6 Fig).

## Discussion

As shown in Results section, when $-\log_{10}(p)$ was evaluated by the LD block unit for the coupling scenario, RAINBOW did not necessarily outperform the other methods. However, if we

## Iteration 48

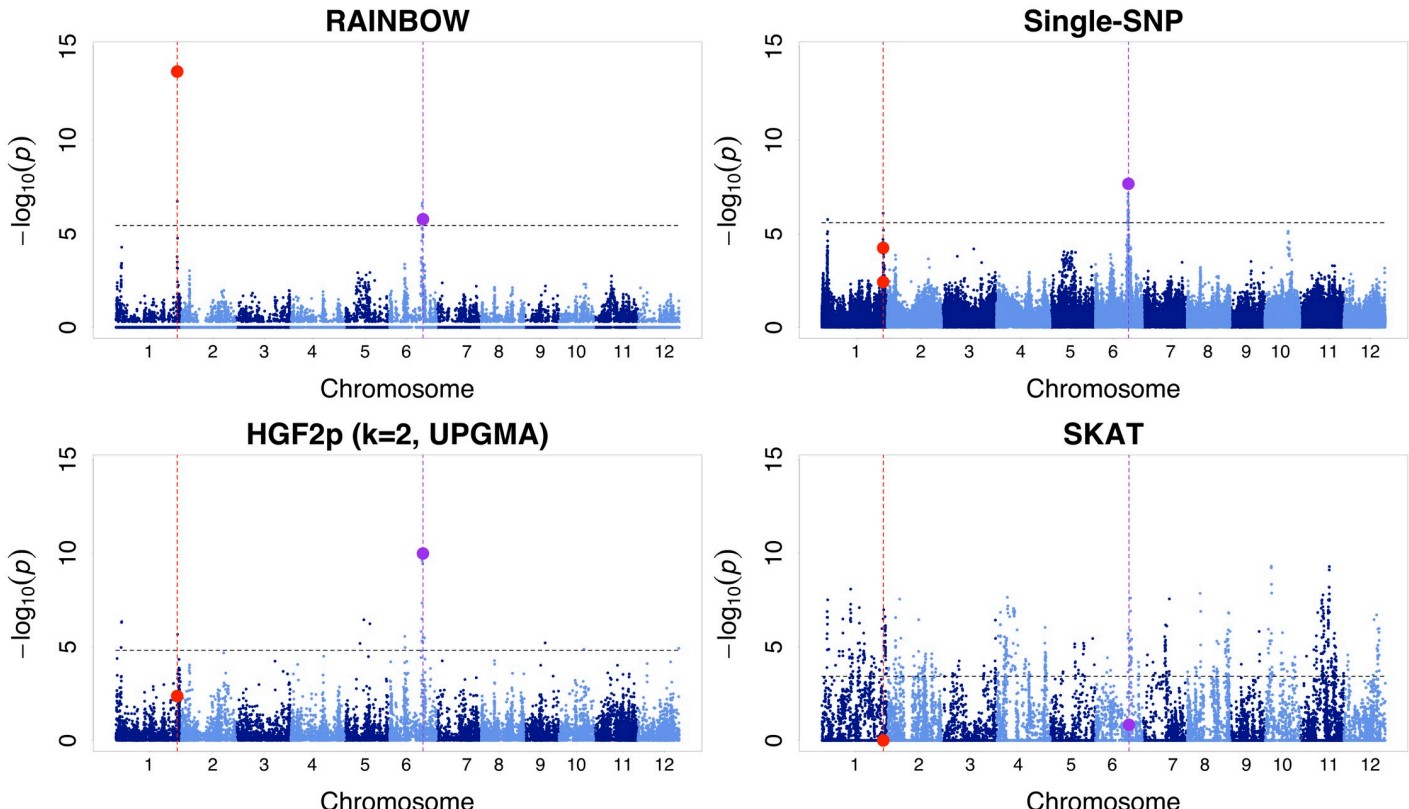

**Fig 4. An example of GWAS results for the repulsion scenario.** Manhattan plots of 4 GWAS methods (RAINBOW, Single-SNP GWAS, HGF (the number of groups is 2, the grouping method is UPGMA), and SKAT) for one simulation result of the Repulsion model. The black horizontal dashed lines represent the thresholds determined by the Benjamini-Hochberg method (FDR = 0.01) for each result of the Repulsion model. The red vertical dashed lines show the positions of QTN1, QTN2, and the purple ones show the position of QTN3. The red points show $-\log_{10}(p)$ of causal SNPs or haplotypes including QTN1 and QTN2, and the purple ones show $-\log_{10}(p)$ of QTN3 or haplotypes including QTN3.

considered the inflation level of each result and evaluated the results with the $-\log_{10}(p_a)$, RAINBOW showed as great a detection power as other methods (Fig 1d), which means RAINBOW succeeded in controlling false positives compared with other haplotype-based GWAS methods. This can also be seen from the fact that the precision of RAINBOW was much higher than the other GWAS methods including single-SNP GWAS (Fig 2).

Moreover, $-\log_{10}(p)$ of RAINBOW was the highest when evaluated by that of the causal SNP/haplotype block itself, which implies that RAINBOW can detect causal haplotype blocks themselves without relying on the LD beyond the scope of the haplotype block. This can also be confirmed by the results that showed that the AUC for the regions around the causal was larger in RAINBOW than in any other methods (Fig 3).

In addition, for the repulsion scenario, RAINBOW succeeded in detecting causal haplotype blocks that were not able to be detected by any other methods including single-SNP GWAS (Fig 4). This result affected other results that RAINBOW outperformed the other methods especially when evaluated by the detection power, recall, precision and F-measure in the repulsion scenario. This fact suggests that RAINBOW is good for detecting the causal haplotype block with multiple causal variants. For example, RAINBOW can be applied to the detection

of genes that have more than one variant. Therefore, for future analysis, RAINBOW can be used for gene-set GWAS (which regards one gene as one SNP-set) by using gene annotation information.

The only drawback of RAINBOW is that the detection power for the causal with small effects (QTN3) was not so high (c,d in S4 Fig). The drawback may be related to the fact that RAINBOW succeeded in detecting QTN1 and QTN2 well. In other words, RAINBOW cannot account for the loci of large effects well when testing other loci, and the loci of relatively small effects may be concealed by these loci of large effects. This drawback, however, can be easily resolved by using methods that condition the loci of large effects, such as composite interval mapping for QTL analysis [59, 60] or a multi-locus mixed model for GWAS [61]. For future analysis, we will implement this function to condition the loci of large effects when testing other loci of small effects.

## Supporting information

**S1 Appendix. Supplementary Note for additional RAINBOW methods.** A faster computational method for the linear kernel and effective testing method for dominance and epistatic effects are mainly described.
(PDF)

**S1 Table. Supplementary table for accession information used in this study.**
(CSV)

**S1 File. Source codes for the R package RAINBOWR.** Including source code and license files for the R package RAINBOWR. Please see "Readme.md" file to start the RAINBOW.
(RAR)

**S1 Fig. Supplementary figure for the flow chart of the simulation framework in tis study.**
(PDF)

**S2 Fig. Supplementary figure for $-\log_{10}(p)$ and $-\log_{10}(p_a)$ of QTN3 for each method.** How to view this figure (including legends and abbreviations) is the same as that of Fig 1.
(PDF)

**S3 Fig. Supplementary figure for recall, precision and F-measure determined by the threshold criterion of Bonferroni correction whose significance level equals to 0.01.** How to view this figure (including legends and abbreviations) is the same as that of Fig 2.
(PDF)

**S4 Fig. Supplementary figure for recall, precision and F-measure of each QTN.** How to view this figure (including legends and abbreviations) is the same as that of Fig 2.
(PDF)

**S5 Fig. Supplementary figure for recall, precision and F-measure of each QTN determined by the threshold criterion of Bonferroni correction whose significance level equals to 0.01.** How to view this figure (including legends and abbreviations) is the same as that of Fig 2.
(PDF)

**S6 Fig. Supplementary figures (6 pages) for the examples of the cases where only RAINBOW succeeded in detecting causals, for the repulsion scenario.** How to view this figure (including legends and abbreviations) is the same as that of Fig 4.
(PDF)

## Acknowledgments

We are grateful to Dr. Ryokei Tanaka and Dr. Shiori Yabe for fruitful discussions, Dr. Motoyuki Ishimori and Mr. Goshi Sasaki for debugging the package, and Mr. Ryusuke Hamazaki for naming the package, RAINBOW.

## Author Contributions

**Conceptualization:** Kosuke Hamazaki, Hiroyoshi Iwata.

**Data curation:** Kosuke Hamazaki.

**Formal analysis:** Kosuke Hamazaki.

**Funding acquisition:** Hiroyoshi Iwata.

**Investigation:** Kosuke Hamazaki.

**Methodology:** Kosuke Hamazaki.

**Project administration:** Hiroyoshi Iwata.

**Resources:** Kosuke Hamazaki.

**Software:** Kosuke Hamazaki.

**Supervision:** Hiroyoshi Iwata.

**Validation:** Kosuke Hamazaki.

**Visualization:** Kosuke Hamazaki.

**Writing – original draft:** Kosuke Hamazaki.

**Writing – review & editing:** Hiroyoshi Iwata.

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
