## [Decision Letter · Decision Letter 0]

11 Nov 2019

Dear Dr Iwata,

Thank you very much for submitting your manuscript 'RAINBOW: Haplotype-based genome-wide association study using a novel SNP-set method' for review by PLOS Computational Biology. Your manuscript has been fully evaluated by the PLOS Computational Biology editorial team and in this case also by independent peer reviewers. The reviewers appreciated the attention to an important problem, but raised some substantial concerns about the manuscript as it currently stands. While your manuscript cannot be accepted in its present form, we are willing to consider a revised version in which the issues raised by the reviewers have been adequately addressed. We cannot, of course, promise publication at that time.

Sincerely,

Mihaela Pertea

Software Editor

PLOS Computational Biology

Mihaela Pertea

Software Editor

PLOS Computational Biology

[LINK]

Reviewer's Responses to Questions

**Comments to the Authors:**

Reviewer #1: This is a very interesting research that attempts to improve rare variants detection of conventional GWAS-SNPs models via haplotypes. The proposed method performs as good as the conventional methods for controlling false positives; however, it shows to outperform the other models detecting causal induced variants that were not identified with the conventional models. Also, one of the advantages of this proposed model is that it does not rely on LD when causal variants are also genotyped.

In general the materials and methods, Results and Discussion sections are well written; however, the abstract and introduction sections needs some improvements. Especially for describing better the scope and implications of the results of the proposed method.

Here a few minor points.

Page 2, lines 6-9. sequencing data is the "3000 rice genomes project" [].

such public data, the conventional GWAS

Page 2, line 29. in false

as in the world collection of rice germplasm used in this

drawback: a weighting

Line 39. which is a computationally

method since it requires

Page 3, lines 47-49. Please rephrase.

Derivations of the equations and model development is ok

Adding a diagram for explaining the proposed simulation scheme would help to understand better the results.

Page 8, line 217. data an material

Discussion was well conducted. Perhaps a conclusions section would be desirable if that is allow in the journal format.

Reviewer #2: Please see attachment for review.

**Have all data underlying the figures and results presented in the manuscript been provided?**

Reviewer #1: Yes

Reviewer #2: Yes

PLOS authors have the option to publish the peer review history of their article (what does this mean?). If published, this will include your full peer review and any attached files.

Reviewer #1: Yes: Diego Jarquin

Reviewer #2: No

---

## [Decision Letter · Decision Letter 1]

18 Jan 2020

Dear Dr. Iwata,

We are pleased to inform you that your manuscript 'RAINBOW: Haplotype-based genome-wide association study using a novel SNP-set method' has been provisionally accepted for publication in PLOS Computational Biology.

Before your manuscript can be formally accepted you will need to complete some formatting changes, which you will receive in a follow up email. A member of our team will be in touch within two working days with a set of requests.

Best regards,

Mihaela Pertea

Software Editor

PLOS Computational Biology

Mihaela Pertea

Software Editor

PLOS Computational Biology

Reviewer's Responses to Questions

**Comments to the Authors:**

Reviewer #1: I have no further comments on the version of the manuscript. All my questions were correctly addressed by the authors.

Reviewer #2: The authors have substantially improved their writing.

Their contribution have tried to address GWAS - an important but difficult problem. The method requires estimating variance components of models from a multi-step approach, which includes estimating weights to scale the estimated variance components from a model with a single random effect.

I am not convinced such an approach is optimal to estimation of variance components directly but acknowledge that the contribution and results are worthy of dissemination. A weakness in the method include that it is not easily extendible (e.g. if there are three random effects then the algorithm needs modification) - perhaps something that authors may like to think about in future developments of their software.

**Have all data underlying the figures and results presented in the manuscript been provided?**

Reviewer #1: Yes

Reviewer #2: Yes

PLOS authors have the option to publish the peer review history of their article (what does this mean?). If published, this will include your full peer review and any attached files.

Reviewer #1: No

Reviewer #2: No

---

## [Editor Report · Acceptance letter]

6 Feb 2020

PCOMPBIOL-D-19-01767R1 

RAINBOW: Haplotype-based genome-wide association study using a novel SNP-set method

Dear Dr Iwata,

I am pleased to inform you that your manuscript has been formally accepted for publication in PLOS Computational Biology. Your manuscript is now with our production department and you will be notified of the publication date in due course.

With kind regards,

Sarah Hammond
